# Advances in Electrostatic Spinning of Polymer Fibers Functionalized with Metal-Based Nanocrystals and Biomedical Applications

**DOI:** 10.3390/molecules27175548

**Published:** 2022-08-29

**Authors:** Haojun Li, Meng Xu, Rui Shi, Aiying Zhang, Jiatao Zhang

**Affiliations:** 1Institute of Medical-Industrial Integration, Beijing Key Laboratory of Structurally Controllable Advanced Functional Materials and Green Applications, School of Materials Science and Engineering, Beijing Institute of Technology, Beijing 100081, China; 2Jishuitan Hospital, Beijing 100035, China; 3Key Laboratory of Medical Molecule Science and Pharmaceutics Engineering, Ministry of Industry and Information Technology, School of Chemistry and Chemical Engineering, Beijing Institute of Technology, Beijing 100081, China

**Keywords:** electrostatic spinning, inorganic nanocrystals, polymer fibers, antimicrobial, biomedical

## Abstract

Considering the metal-based nanocrystal (NC) hierarchical structure requirements in many real applications, starting from basic synthesis principles of electrostatic spinning technology, the formation of functionalized fibrous materials with inorganic metallic and semiconductor nanocrystalline materials by electrostatic spinning synthesis technology in recent years was reviewed. Several typical electrostatic spinning synthesis methods for nanocrystalline materials in polymers are presented. Finally, the specific applications and perspectives of such electrostatic spun nanofibers in the biomedical field are reviewed in terms of antimicrobial fibers, biosensing and so on.

## 1. Introduction

In recent decades, electrostatic spinning has received more and more attention as an important technology for the preparation of micro/nanomaterials due to the rapid demand for fiber-based applications. Electrostatic spinning is a process in which a polymer solution is stretched into nanofibers by applying a high-voltage electric field. The concept of electrostatic spinning was conceived as early as 1600 by Gilbert, who observed in a study that polymer solutions could form conical droplets under an electric field [1]. In 1882, Taylor published a series of seminal papers stating that as the electric field strength increased to a critical level, the spherical droplet would gradually evolve into a cone [2] (now commonly referred to as a “Taylor cone”) and emit a liquid jet. The application of the electrojet process to the preparation of fibers eventually evolved into electrostatic spinning technology. After 1990, Reneker’s group and Rutledge’s group conducted more in-depth research on electrostatic spinning technology and applications [3,4]. Recently, more and more attention has been paid to electrostatic spinning by using new materials to fabricate composite materials and ceramic nanofibers, due to the fact that new applications in electrostatic spinning fibers are used in in soft electronic devices [5,6,7], biomedicine [8,9], energy harvesting, conversion and storage [10,11]. The number of publications on electrospinning keeps increasing, as shown in Figure 1A, which also indicates the new opportunities in the development of electrostatic spinning.

The ability to encapsulate hydrophobic and hydrophilic compounds and functional nanomaterials directly into fibers is a major advantage in electrostatic spinning technology [12,13,14]. This is because electrostatic spinning can be performed under relatively mild environmental conditions, which could retain the activity of the loaded substances during the forming process, making them more suitable for encapsulating active nanomaterials that are sensitive to heat than other conventional processing strategies. As for bio-applications, it has been demonstrated that cells could also be processed by electrostatic spinning without loss of their activities [15,16,17]. In addition, electrospun fibers, with tunable diameters from submicron to nanometer level and high specific surface area, can also facilitate the dispersion of their loaded compounds into the surrounding medium, thereby controlling the release of active substances (e.g., drugs) [18,19,20,21]. In addition, electrospun fibers can mimic the microstructure of the human extracellular matrix, thus, greatly improving the biocompatibility of the material and making it more stable for the loaded bio-nanomaterials and drugs [22].

Due to the aforementioned properties, electrospun fibers have promising applications in the field of biomedical materials, especially in the controlled release of compounds in drug delivery [23,24], scaffolds in tissue engineering [25,26] and wound dressings [27,28]. The number of publications on the bio-application of electrospinning technology also keeps increasing in the most recent 20 years, as shown in Figure 1B. In this work, we summarize the principles of electrostatic spinning, with emphasis on the two main methods of loading inorganic NCs onto electrospun membranes and their advantages and limitations. Then, we review the applications of electrospun fiber membranes loaded with inorganic NCs in the field of antibacterial and biosensing directions, especially publications reported in the last 10 years. Finally, we discuss the challenges and opportunities of electrostatic spinning, especially for nanosynthesis.

## 2. The Principle of Electrostatic Spinning Technology

### 2.1. Principle of Electrostatic Spinning

A schematic diagram of the electrostatic spinning for preparing nanofibers is shown in Figure 2, including strategies, such as directly mixing, in situ growth and assembly of inorganic NCs. The equipment of electrostatic spinning generally consists of three parts: the spinneret, the high-voltage power supply and the receiving device [29]. In general, the electrostatic spinning process can be divided into the following four steps: (i) the polymer solution forms a Taylor cone under an electric field; (ii) the charged jet extends along a straight line under the electric field; (iii) the jet becomes finer under the electric field and the electric bending instability (also called agitation instability) increases; and (iv) the jet condenses into solid fibers and is collected on a grounded collector [30]. Among them, the formation of the Taylor cone is the most critical step in this process, which determines the quality of the fibers. In the electrospinning process, the metal whiskers are easily formed. The static polarization of the wire in the electric field brings about an energy gain, resulting in a metal whisker that appears as a hair-like protrusion on the surface of some metal [31]. In spite of the potential weakness caused by metal whiskers in the electronic industry, the mechanisms in the formation of metal whiskers still need further investigation.

### 2.2. Main Factors Influencing the Formation of Electrostatic Spinning Nanofibers

The formation of electrospun fibers and the control of their diameters depend largely on the processing parameters, including the polymer solution concentration, applied voltage, liquid flow rates, distance between the spinneret tip and the collector, etc.

#### 2.2.1. Polymer Solution Concentration

As the concentration of the polymer solution increases, the viscosity and surface tension of the solution also increase, resulting in a less stretchable solution, and the jets formed in the electric field become less likely to split and become finer, resulting in a larger diameter for the collected fibers. Thus, under the same conditions, the diameter of nanofibers increases with an increase in polymer concentration. When the concentration of the solution is too low, the viscosity of the solution is also low, which could be easy electrostatic atomization, resulting in the presence of a large number of string beads in the fiber [32]. Therefore, in order to obtain the ideal electrospun fibers, it is necessary to find the most suitable polymer concentration for electrospinning.

#### 2.2.2. Liquid Flow Rate and Receiving Distance

The injection rate also affects the structures of fibers by influencing the formation of Taylor cones. In general, as the injection rates increase, the fiber diameter also increases. Typically, the jet needs a long enough distance to extend and coagulate before forming solid fibers. Normally, the fibers become finer as the receiving distance increases. After a certain distance, the fibers will no longer become finer due to the solidification of the jet [30].

#### 2.2.3. Electric Field Voltage

A static high-voltage direct current is usually applied to the spinneret to generate an electric field. The magnitude of the voltage determines the amount of charge carried by the jet and the strength of the electric field. Applying a high voltage usually tends to form thinner fibers [33], while it might also cause more liquid to be injected, resulting in larger fiber diameters [34].

## 3. Noble Metal, Semiconductor Nanocrystalline Materials and Their Electrostatic Spinning into Polymers

### 3.1. Noble Metal, Semiconductor Nanocrystalline Materials

With the rapid development of nanotechnology and science, colloidal inorganic NCs with different morphologies, sizes and functions (Figure 3) have been investigated, including noble metal nanoparticles (NPs) [35,36,37], semiconducting metal oxides [38,39], single-atom catalysts [40,41,42,43], hybrid NCs [44,45,46], noble metal nanoclusters [47] and semiconductor quantum dots (QDs) [48,49,50,51,52], etc. These kinds of nanomaterials are widely used in the fields of energy [53,54], catalysis [55,56], sensing [57], electronic information [58], optoelectronic devices [59], biomedicine and imaging [60,61,62] because of their unique optical, electronic, magnetic, thermal and mechanical properties. Dispersing nanomaterials into a bulk matrix, such as large-scale polymer fibers, with low concentrations but well-maintained intrinsic nano effect and properties, has been regarded as a promising strategy to further explore their potential applications in our daily life.

### 3.2. Preparation of Nanocrystalline Functionalized Electrospun Composite Fibers

There are many strategies to assemble NCs into electrospun fibers and direct mixing and in situ growth are two most representative strategies. By combining inorganic NCs with electrospun nanofibers, the stability of many kinds of such NCs can be effectively improved. Moreover, their intrinsic properties could be well maintained. In addition, the advantages and disadvantages of each of these two methods are listed in Table 1.

#### 3.2.1. Direct Mixing with NCs in Polymer Precursors

Due to the properties of small size effect and high surface energy, NCs could be aggregated in long-term practical applications. However, when mixing these inorganic NCs with the solutions of electrospinning polymers, NCs would be coordinated and stabilized by the surficial ligands of electrospun polymers, which could inhibit the aggregations of NCs. At present, many kinds of inorganic NCs, such as noble metal, metal oxygen/sulfide and semiconductor NCs, have been directly mixed and blended in electrospun nanofibers [63,64,65,66]. For example, El-Hefnawy et al. reported the synthesis of Ag NC dispersed polymer fibers [67]. The fabricated Ag NCs were made into a monodisperse form with a diameter of no more than 6 nm. Prior to the electrospinning process, they added different volumes of Ag NCs dropwise into the polymer mixture solution to obtain nanofiber sheets containing different concentrations of Ag NCs. They found that the fibers containing Ag NCs were uniform and the diameter of the fibers could be tunable by increasing the concentration of Ag NCs.

Similarly, Manjumeena et al. reported the synthesis of Au NCs deposited on polyvinyl alcohol (PVA) nanofibers enabled by the dispersion of PVA and Au NCs in distilled water by controlling the corresponding electrospun conditions [68]. Electrospun nanofibers loaded with Au NCs were obtained and the scanning electron microscope (SEM) and high-resolution transmission electron microscope (HRTEM) results indicated that Au NCs were located on the surface of the electrospun fiber. They concluded that the PVA could become more hydrophilic after being loaded with a small amount of Au NPs. Another study, by Li et al., reported the synthesis of Fe_3_O_4_-modified electrospun fibers [69]. They found that at high voltages, Fe_3_O_4_ could improve the arrangement of fibers compared to pristine electrospun fibers. Two-dimensional NCs could also be deposited on nanofibers, as reported by Somia et al. (Figure 4) [70]. Cellulose nanocrystalline-ZnO (CNC-ZnO) hybrids were obtained using the hydrothermal method followed by dispersion in chloroform/DMF mixed solvent with additionally dissolved pl (PHBV). PHBV/CNC-ZnO composite nanofibers were successfully obtained through the electrospinning strategy. It was found that after the combination of sheet-like CNC-ZnO and PHBV, the nucleation density, overall crystallinity and crystallinity in PHBV composite nanofibers were significantly improved and their thermal degradation temperature also increased.

#### 3.2.2. In Situ Growth of NCs on Electrospun Fibers

In situ growth of NCs on electrospun fibers is another efficient strategy to achieve NC-modified nanofibers. Generally, the precursor solution of the metal salt is dissolved in the electrospinning solution followed by using light, electricity, heat, chemical reduction and other methods to trigger the reduction in and oxidation of metal ions in the electrospun solution or electrospinning fiber. As shown in Figure 5, Song Lin et al. reported two in situ growth methods to obtain Ag NC-loaded PVA nanofiber pads. The first method was to reflux the AgNO_3_-soluble PVA solution at 105 °C for 1 h, resulting in Ag NCs being generated in this process, and then the nanofibers were obtained by the electrospinning process. The second method was to dissolve PVA in deionized water to obtain a viscous solution and then add AgNO_3_ as an electrospinning solution. After the end of electrospinning, Ag NCs could be formed inside the nanofibers under full ultraviolet (UV) lamp illumination. Among them, the size and yield of doped Ag NCs can be adjusted by controlling preheating treatment or UV irradiation [71]. Soon et al. prepared polyacrylonitrile (PAN) electrospun fiber membranes supporting Pt NTHFPs (NPs) by in situ calcination [72]. The PAN electrospun fibers were prepared first followed by immersion in Pt(acac)_2_ acetone solution to load Pt. After heat treatment in an inert atmosphere, the nanofibers loaded with Pt NPs were carbonized at high temperatures. In another representative study by Dakota et al., Ag NP-modified polycaprolactone (PCL) nanofibers were fabricated by in situ plasma treatment [73]. First, the PCL and AgNO_3_ were dissolved in acetone followed by electrospinning under appropriate conditions. The electrospun fibers were then treated with air plasma, which could be a simple and effective method to generate Ag NPs on fiber membranes.

Similarly, Wang et al. dissolved polylactic acid (PLA)/PCL and Cu_2_S NPs in a N-N dimethylformamide/tetrahydrofuran (DMF/THF) solvent mixture forming Cu_2_S NC-mixed polymer fibers, as shown in Figure 6A [74]. It was shown that these fiber films exhibited excellent and controllable photothermal properties under NP (NIR) irradiation, showing great promise in tumor-induced wound healing applications. Compared to single-component-modified nanofibers, Yu’s group reported the fabrication of assembled binary component NC-modified nanofibers by embedding Au nanorods (NRs) and silver nanowire (Ag NW) assemblies into PVA electrospun nanofibers to improve the stability of Au NRs/Ag NWs, as shown in Figure 6B [75]. When using a woven-structured copper mesh as the receiver device, the Au NRs/Ag NW assemblies were mostly distributed in a directional manner within the electrospun fibers. Furthermore, because of the polarization effect of the Ag NP-polymer solution under the high-voltage power supply, they distributed the dimer and small aggregates of Ag NPs directionally inside the PVA fiber and the composite fiber finally produced a more desirable surface-enhanced Raman scattering (SERS) effect [76].

As for the synthesis of semiconductor NC-modified nanofibers, as shown in Figure 7A, Kampara et al. used an in situ calcination strategy to obtain a PVA electrospun fiber membrane loaded with CdO semiconductor NCs [77]. The precursor to synthesize CdO NCs was cadmium acetate dihydrate. After the initial electrospinning, the original nanofiber membranes were transferred to the muffle furnace and calcined at high temperature to obtain the product. As shown in Figure 7B, Kamal et al. prepared PLA/titanium dioxide hybrid nanofibers using the in situ hydrothermal method. The coated fibers were obtained by combining electrospinning and electrospraying techniques. The electrospinning solution was prepared by dissolving PLA in a mixed solvent of dichloromethane/methanol. A mixture of tetraisopropoxide Ti(O-iPr)_4_ (TIP), ethanol and hydrochloric acid was used for electrospraying, as a precursor to Ti. The obtained coating fibers were sufficiently dried under vacuum and then transferred to an autoclave for hydrothermal treatment and the Ti precursors were finally converted into TiO_2_ NPs [78].

## 4. Functionalized Electrostatic Spinning Composite Fibers for Biomedical Applications

By virtue of the aforementioned advantages of electrospun nanofibers, such as nanoscale size, high porosity and large specific surface area, their potential applications have been widely studied in many fields [5,6,7,8,9,10,11]. Specifically, nontoxic electrospun nanofibers were regarded as promising candidates for biomedicines [23,24,25,26,27,28], with adjustable properties, such as drug release, wound dressing, tissue engineering and trauma repair.

### 4.1. Electrostatic Spun Nano-Antimicrobial Fibers

Many kinds of noble metals and oxides of some metals can exhibit certain antimicrobial properties [79,80,81]. Ag NPs are some of the most typical antibacterial NCs by virtue of the advantages of adjustable size, excellent antibacterial effect, continuous antibacterial effect, etc., which exhibits a wide range of applications in the field of antibacterial biology. As shown in Figure 8A, Yan et al. prepared PVA nanofibers loaded with Ag NPs using an in situ hydrothermal assay [82]. They assessed the bactericidal properties of pure PVA and Ag NPs/PVA through turbidity and absorption methods for E. coli and S. aureus. Their results indicated that the latter exhibited more excellent antibacterial properties. The amount of fiber-loaded Ag NPs was also controlled by adjusting the concentration of AgNO_3_, which indicated that samples with a concentration of AgNO_3_ at 0.066 mol/L had the highest antimicrobial rates against E. coli and S. aureus, at 98% and 99%, respectively. By direct mixing, Erick et al. incorporated Ag NPs into PCL electrospun nanofibers to study their antimicrobial properties. They demonstrated the antibacterial activity of fiber scaffolds by agar diffusion and the results indicated that the antibacterial activity of fiber scaffolds on S. aureus, E. coli, K. pneumoniae and P. aeruginosa was directly proportional to the concentration of Ag NPs. Compared with Gram-negative bacteria (E. coli, P. aeruginosa and K. pneumoniae), Gram-positive strains (S. Aureus, S. mutans, B. subtilis) were more sensitive to PLA-Ag NPs nanofibers [83]. Reza et al. studied the wound healing effects of compound nanofibers embedded with Ag NPs. The antibacterial activity of the product against E. coli, P. aeruginosa and S. aureus was studied in vitro and the results indicated that the higher the silver content, the better the antibacterial effect. The product was tested for cytotoxicity in vitro using the MTT assay and the results showed that the fiber scaffold was nontoxic and had good biocompatibility. They used nanofiber pads on wounds caused by resection of white rabbits in New Zealand to study their effects as wound dressings. Silver-containing nanofiber membranes showed good healing properties compared to Ag-free polyvinylalcohol/polyvinylpyrrolidone/pectin/mafenide acetate (PVA/PVP/PEC/MF) nanofibers and obtained the best wound healing effect when the composition ratio of Ag NPs/PVA/PVP/PEC/MF was 0.7:91.8:2.5:2.5 wt% [84]. As shown in Figure 8B, Qian et al. developed a novel Ag-modified/collagen-coated electrospun p/polycaprolactone (PLGA/PCL) scaffold (PP-pDA-Ag-COL) with improved antimicrobial and osteogenic properties [85]. The scaffold was generated by electrospinning a basic PLGA/PCL matrix, followed by Ag NPs impregnation via in situ reduction, polydopamine coating and then coating by collagen I. The three intermediate materials involved in the fabrication of the scaffolds, namely, PLGA/PCL (PP), PLGA/PCL-polydopamine (PP-pDA) and PLGA/PCL-polydopamine-Ag (PP-pDA-Ag), were used as control scaffolds. There was a wider antibacterial zone associated in PP-pDA-Ag-COL and PP-pDA-Ag scaffolds versus control scaffolds (*p* < 0.05) and bacterial fluorescence was reduced on the Ag-modified scaffolds after 24 h inoculation against Staphylococcus aureus and Streptococcus mutans. In a mouse periodontal disease model, the PP-pDA-Ag-COL scaffold enhanced alveolar bone regeneration (31.8%) and was effective for periodontitis treatment. These results demonstrate that this novel PP-pDA-Ag-COL scaffold enhanced biocompatibility and osteogenic and antibacterial properties.

TiO_2_ NCs, as an inorganic antibacterial agent, especially under light irradiation, have also gradually received attention as a promising nano-antibacterial material by virtue of their advantages of high stability, nontoxicity and easily manipulated properties. Pant et al. prepared TiO_2_-containing nylon-6 nanofibers using electrospinning technology and experimentally demonstrated that the nanofibers had good antibacterial properties [86]. Toniatto et al. reported the synthesis of TiO_2_-modified PLA by direct mixing [87]. The prepared composite nanofibers were tested using thiazole blue colorimetry (MTT method) and the results showed that the composite nanofibers had no significant cytotoxicity. Through the evaluation of antibacterial experiments, the composite nanofibers with a content of 5 wt% show strong bactericidal properties.

In addition to the above materials being widely used in the field of antibacterial biology, nanomaterials, such as Au NPs [88], Se NPs [89] and ZnO NCs [90], also exhibit excellent performance in this field.

### 4.2. Biosensing Applications

Biosensing is an important branch of chemical sensing, which has been applied for the detection of small biological molecules, enzymes, nucleic acids, disease markers, cells, bacteria, etc. As for different biological reactions, designing and constructing suitable NC/electrospun composite fiber membranes are important for biosensing applications.

In a recent study reported by Beak et al., Cu nanoflower-modified Au NP-graphene oxide (GO) nanofibers were synthesized as electrochemical biosensors for glucose detection using a novel electrospinning method [91]. Electrochemical experiments showed that Cu-nanoflower@AuNPs-GO nanofibers have the advantages of high sensitivity, low detection limit and good reproducibility and selectivity in detecting glucose. In addition to the special catalytic properties, metal oxides could also facilitate electron transfer, which could provide a more friendly electroactive surface, thus, enabling the direct transfer of electrons to the electrode. For example, Li et al. prepared uniformly dispersed Pd NPs anchored on CuO nanofibers through the electrostatic spinning method, which were used to construct enzyme-free glucose sensors, with the advantages of fast response, high sensitivity and low detection limit [92]. Liu et al. also used ZnO nanostructures as an immobilized substrate for an enzyme glucose sensor and immobilized glucose oxidase on it, thus, enabling it to directly undergo electron transfer with the electrode and exhibit high catalytic activity, with a wide linear range and high sensitivity [93]. Perovskite NCs with high optoelectronic properties were also applied for biosensing applications. For example, Wang et al. prepared monolithic superhydrophobic polystyrene fiber membranes encapsulated with CsPbBr_3_ quantums (CPBQD) by one-step electrospinning [94]. The fiber membrane composite coupled CPBQD with a polystyrene (PS) matrix and showed high quantum yield (~91%), narrow half-peak width (~16 nm) and ~100% fluorescence retention after 10 days of exposure to water. Thanks to the excellent optical properties of CPBQD, an ultra-low detection limit of 0.01 ppm was obtained for Rhodamine 6G (R6G) detection and the HRTEM (FRET) efficiency was calculated as 18.80% at 1 ppm R6G in aqueous solution.

Hybrid materials composed of polymer nanofibers and plasmonic noble metal NCs have also developed significantly in recent years for biosensing. For example, as shown in Figure 9, Yang et al. designed a plasma-independent substrate consisting of Ag NPs supported on PAN electrospun nanofiber membranes as a bacterial-detection sensor [95]. The substrate exhibited highly sensitive SERS performance for bacterial identification in the absence of specific bacterial-aptamer coupling. The substrate exhibited good homogeneity of SERS response to bacterial organelles. The antimicrobial properties were also evaluated, which indicated that Ag@PAN nanofiber mats have good antimicrobial properties against both Escherichia coli and Staphylococcus aureus. Anitha et al. synthesized a composite nanofiber membrane loaded with Au NPs by a simple direct mixing method for the detection of H_2_O_2_ [96]. By virtue of the uniform distribution and large surface area of the Au NPs in the nanofibers, the Au NP-composite electrodes enabled greatly improved electrochemical properties, compared to Au NP-free composite electrodes. When they were employed as reservoirs for immobilizing horseradish peroxidase, reliable and sensitive electrochemical detection by the enzyme reaction was achieved. Their experimental results demonstrated that the detection sensitivity to H_2_O_2_ could be an order of magnitude higher than other previously reported electrochemical sensors.

### 4.3. Other Applications

In addition to antimicrobial applications and biosensing, NC-loaded electrospun fiber membranes were also applied in other biomedical fields. For example, Ming et al. synthesized an electrospun fiber membrane loaded with Au NRs for photothermal treatment of cancer [97]. This strategy not only utilizes the excellent photothermal properties of Au NRs to selectively kill cancer cells, but also utilizes widely used biodegradable electrospinning membranes as Au NRs carriers and surgical recovery materials. Polyethylene glycol (PEG)-modified Au NRs are embedded in an electrospun fiber membrane consisting of PLGA and PLA-b-PEG. After incubation with the cells in the cell culture medium, the PEG-Au NRs were released from the membrane and taken up by cancer cells, allowing the generation of heat upon NIR irradiation to induce cancer cell death (as shown in Figure 10). For another example, biomaterial-based scaffolds fabricated using the electrospinning technique are promising platforms for bone tissue engineering. In the study of Huang et al., citrate-stabilized Au NPs were encapsulated into polyvinylpyrrolidone/ethylcellulose (P/E) scaffolds fabricated by the coaxial electrospinning technique [98]. The results showed that Au NPs were successfully wrapped in electrospinning brackets and the addition hardly affected fiber morphology, but improved porosity and mechanical properties. In vitro studies revealed that Au NP-incorporated electrospun scaffolds showed excellent biocompatibility and osteogenic bioactivities, wherein the alkaline phosphatase activity, mineralized nodule formation and the osteogenic-related genes expression were enhanced in Ag NP-incorporated electrospun scaffolds compared to the neat P/E electrospun nanofibers. Then, the Ag NP-incorporated electrospun scaffolds were surgically implanted into the defect area of the rat skull bone to test their in vivo bone repairing effect. It was observed that Ag NP-incorporated scaffolds rapidly accelerated bone regeneration in vivo.

## 5. Conclusions

Considering the huge requirements in applications of flexible electronic devices, biomedicine and energy harvesting and conversion, etc., the organization of inorganic NCs as building blocks coordinated into hierarchical cross-dimensional and cross-size micro/nano matrix with maintained high performances would be attractive. In this review, taking advantage of electrostatic spinning technologies induced polymer fibers, the research progress of inorganic NCs/polymer fibers synthesis and biomedical applications were reviewed. As mentioned above, electrostatic spinning, as a kind of simple but effective fiber production technology, has been widely used in medical releasing, biosensing and other fields. However, inorganic NC/polymer fiber composites still have many challenges to explore, even in biomedical fields, as illustrated in Figure 11.

(1) Though NC-modified nanofibers have exhibited high potential in many applications, how to modulate the depositing position of NCs is still challenging. Based on the reported strategies, a portion of NCs could be located inside the nanofibers, which could restrain their access to reactive molecules for efficient catalysis or biomedical applications. Hence, efficient strategies to precisely modulate the depositing sites of NCs on nanofibers could be helpful to further explore their applications [99,100].

(2) Compared to single-component modified nanofibers, composites consisting of an electrospun fiber membrane with multicomponent NCs could be developed for more biomedical applications, such as wound dressing applications [101]. Though there are several related papers, the mechanism of synergistic and potential coupling effects between the different introduced NCs within the electrospun fibers on the catalytic, optical and biocompatible properties should be further investigated, which could play an important role in designing reasonable multi-functional NCs/electrospun composite fiber membranes.

(3) Functional NC-modified electrospun fibers, incorporated with sensing and therapeutic capabilities, could be a potential approach in the development of personalized healthcare [102]. For example, "Electronic skin" has become a hotspot in the field of flexible and wearable electronics. Functional NC-modified electrospun composite fibers could be candidates to realize electronic devices and systems with skin-like properties and functions [103,104].

(4) Hybrid NCs, such as Au/semiconductors hetero-NCs with plasmon enhancement, could also exhibit unique biomedical applications when coupled with nanofibers, such as multi-level enhanced chemodynamic, sonodynamic and therapy applications, based on their sonosensitizers and photodynamic functionalities [49,50,51,52,105,106,107,108,109]. It is believed that with the further development of multi-functional inorganic NC/polymer fibers, electrostatic spinning technology will become one of the most widely used technologies in the medical field.

## Figures and Tables

**Figure 1 molecules-27-05548-f001:**
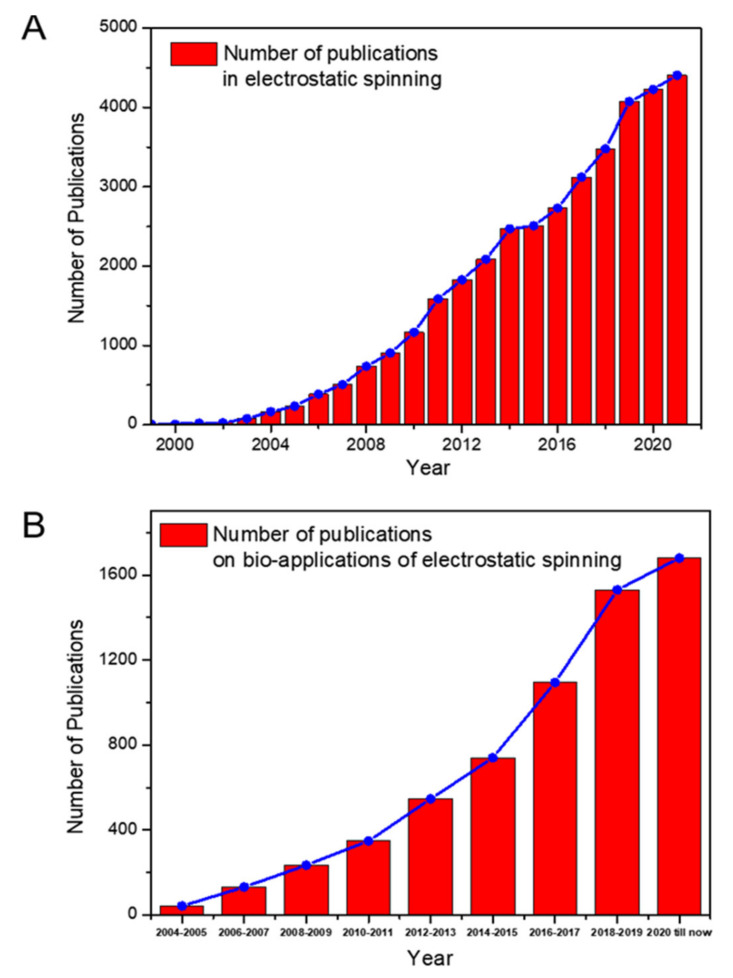
Number of publications on (**A**) electrostatic spinning and (**B**) bio-applications of electrostatic spinning indexed by Web of Science.

**Figure 2 molecules-27-05548-f002:**
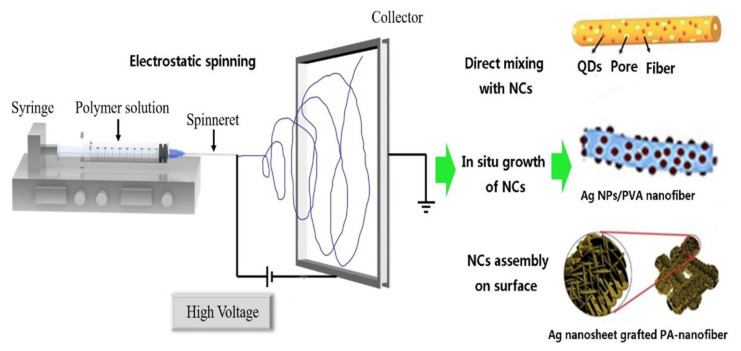
Schematic diagram of nanofibers electrostatic spinning and their direct mixing, in situ growth and assembly of inorganic NCs.

**Figure 3 molecules-27-05548-f003:**
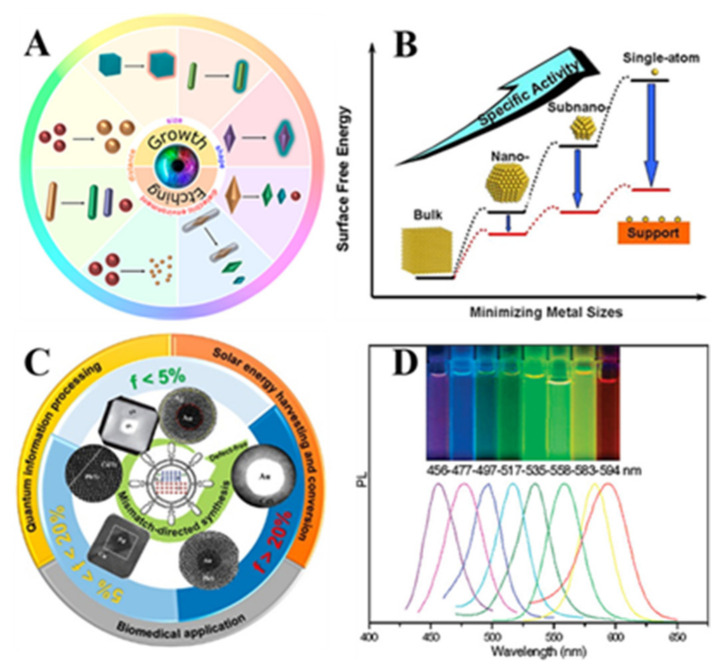
Various nanocrystalline materials. (**A**) Various shapes of noble metal NCs [40]. Copyright 2019 American Chemical Society. (**B**) Subnano- to single-atom catalysts [48]. Copyright 2013 American Chemical Society. (**C**) Hybrid NCs [49]. Copyright 2020 American Chemical Society. (**D**) QDs [57]. Copyright 2008 American Chemical Society.

**Figure 4 molecules-27-05548-f004:**
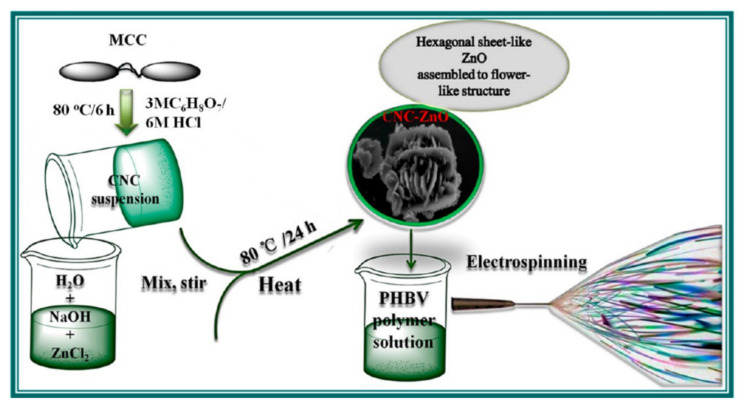
Schematic illustration of possible experimental preparation procedure of sheet-like CNC-ZnO nanohybrids and their electrospinning process [69]. Copyright 2018 American Chemical Society.

**Figure 5 molecules-27-05548-f005:**
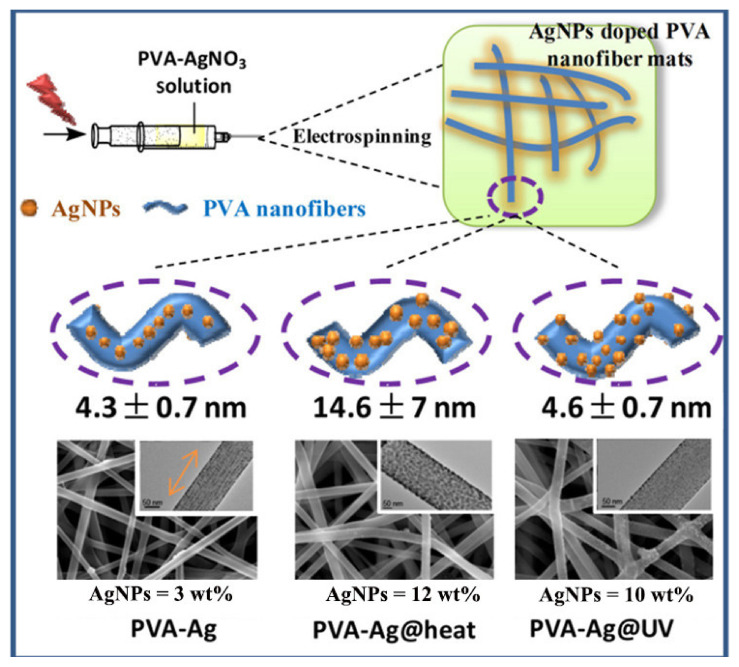
Overview of electrospinning process of Ag NP-doped PVA nanofiber mats under different prepared conditions [70]. Copyright 2022 Dove Press Ltd.

**Figure 6 molecules-27-05548-f006:**
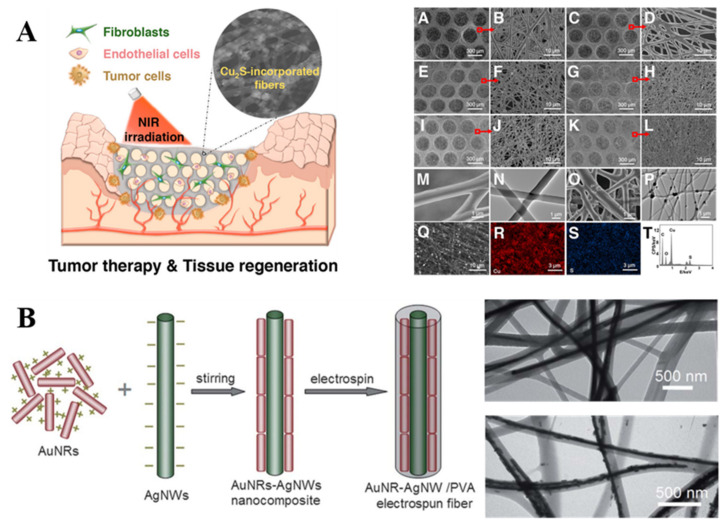
Preparation of nanocrystalline/electrospun composite fibers by direct mixing method. (**A**) Cu2S-incorporated PLA/PCL fiber membrane [74]; Copyright 2017 American Chemical Society. (**B**) Au NR-Ag NWs/PVA electrospun fibers [75]. Copyright 2012 The Royal Society of Chemistry.

**Figure 7 molecules-27-05548-f007:**
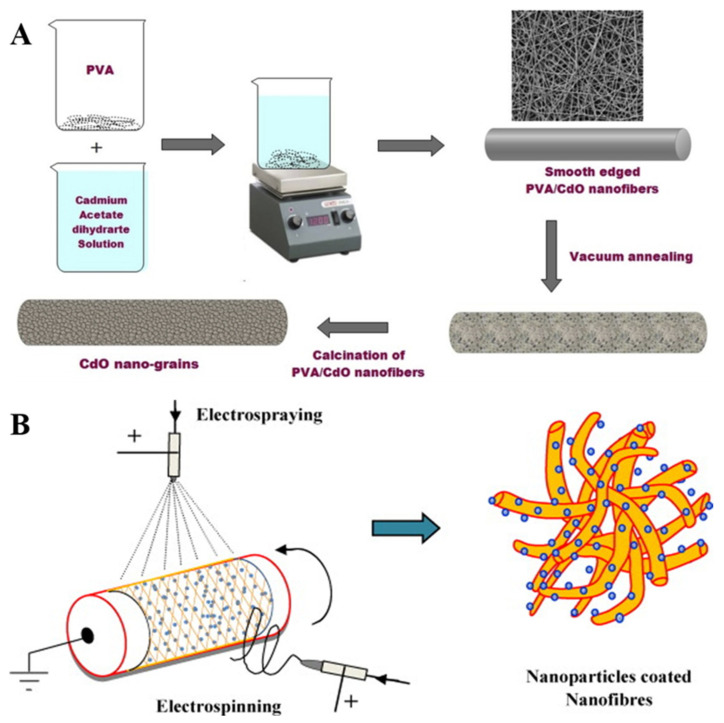
(**A**) Experimental procedure for electrospinning of CdO nanograins [77]. Copyright 2020 Elsevier Ltd. (**B**) Schematic diagram nanoparticle coating on nanofibers using electrospinning and electrospraying [78]. Copyright 2012 Elsevier B.V.

**Figure 8 molecules-27-05548-f008:**
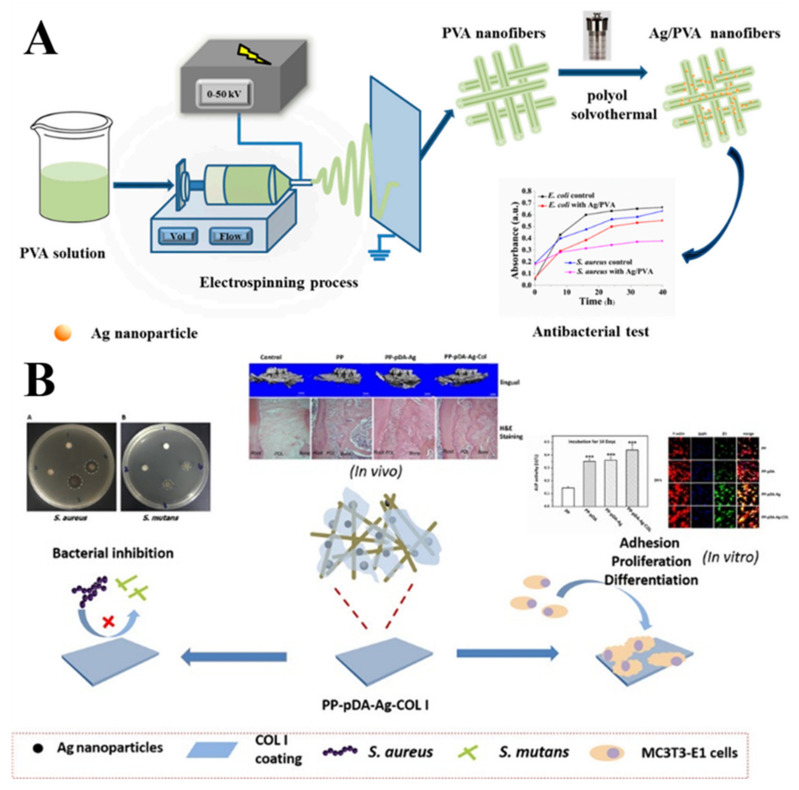
(**A**) Schematic illustration of fabrication and antibacterial test for Ag/PVA composite nanofibers through the electrospinning and solvothermal methods [82]. Copyright 2020 by the authors. (**B**) Triple PLGA/PCL scaffold modification including silver impregnation, collagen coating and electrospinning significantly improve biocompatibility, antimicrobial and osteogenic properties for orofacial tissue regeneration [85]. Copyright 2019 American Chemical Society.

**Figure 9 molecules-27-05548-f009:**
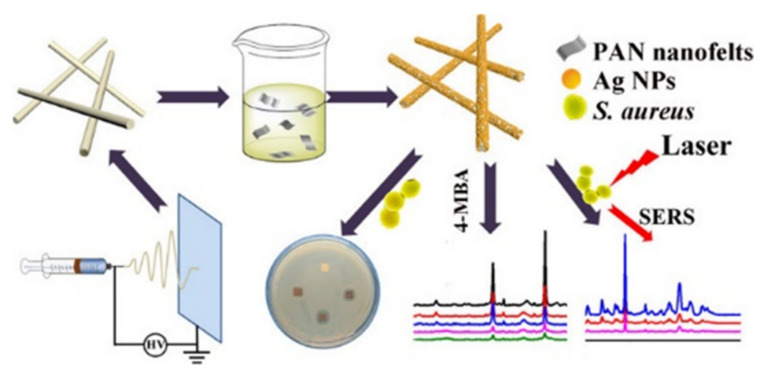
A simple electrostatic spinning technique to prepare Ag@PAN nanofiber membranes for bacterial detection [95]. Copyright 2020 American Chemical Society.

**Figure 10 molecules-27-05548-f010:**
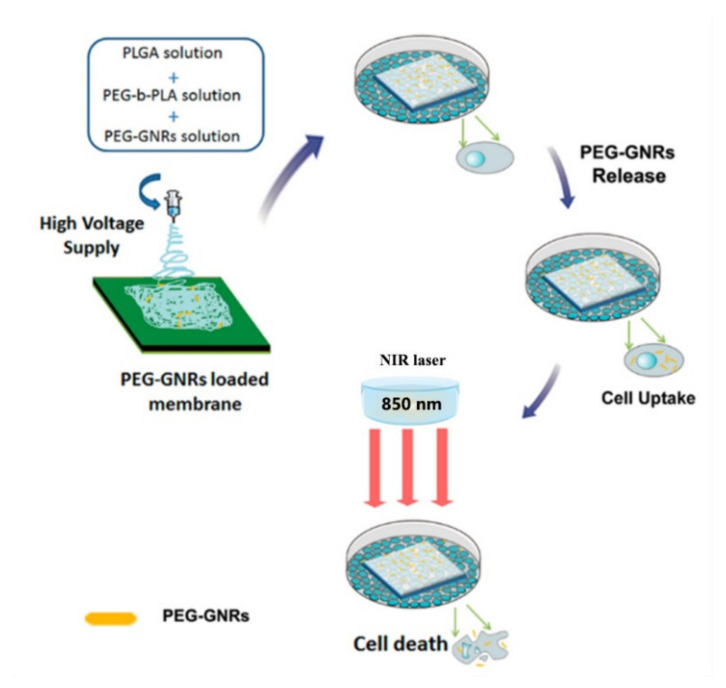
Schematic illustration depicting the strategy of using PEG-GNRs membrane for the photothermal therapy of cancer cells in vitro [97]. Copyright 2014 American Chemical Society.

**Figure 11 molecules-27-05548-f011:**
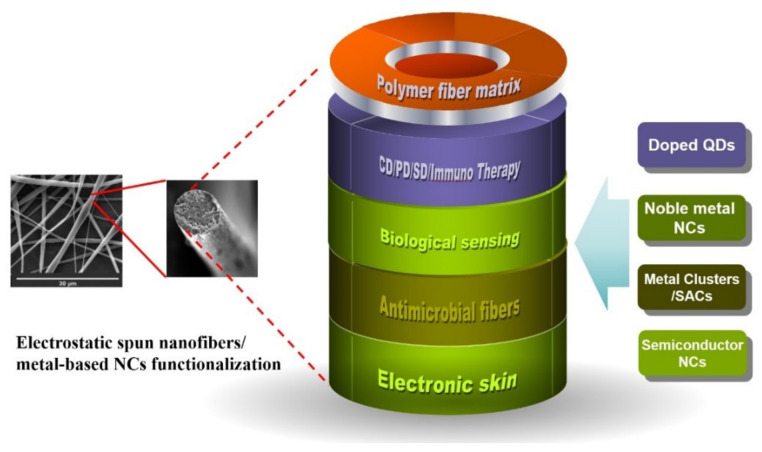
The outlook of metal-based NC functionalization in electrostatic spun nanofibers for extended applications.

**Table 1 molecules-27-05548-t001:** Advantages and disadvantages of direct mixing method and in situ growth method.

Methods	Advantages	Disadvantages
Direct mixing method	(i) Faster and simpler than other compared methods; (ii) Particle sizes and categories depending on pre-synthesized NCs.	(i) Easy to aggregation; (ii) Post-treatment process needed (purification, extractions, etc.); (iii) Lacking size homogeneity in dense matrices; (iv) Restrained connection between NCs and fibers.
In situ growth method	(i) Easy to perform; (ii) Not necessarily extra time in polymeric solution preparation; (iii) No additional solvents required; (iv) Adjustable particle size determined by precursors.	(i) Multi-step reaction; (ii) Additional post-processing time; (iii) Not applicable to all NCs.

## Data Availability

Not applicable.

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
