# Peer review of "Advances in Electrostatic Spinning of Polymer Fibers Functionalized with Metal-Based Nanocrystals and Biomedical Applications"

_molecules, 2022, doi:10.3390/molecules27175548_

Round 1
Reviewer 1 Report
The manuscript is plenty of acronyms don´t previously defined.
Authors refer to 16 and 19 th centuries for electrospinning fundamentals. The metal wiskers and so on fundamentals would be wellcome at this "review". Otherwise, a review in a such complex issue with only 17 pages, looks as very poor. Good (not excellent indeed) as a starting postgraduate work, but very very poor as a true and useful review.
Author Response
We would like to thank Reviewer 1 for his/her kind suggestions. The details of response please see the attachment.

Reviewer 2 Report
This paper shows a good review comparing and summarizing the use of inorganic metallic and semiconductor nano-crystalline materials by electrostatic spinning synthesis technology. There are some issues that need to address:
- I believe this review would benefit from a table that could compare and provide an overview of the discussed approaches. The table should include the advantages and limitations of each approach as well as findings.
- The language of the paper needs to be improved. There are some grammatical errors, carefully check the whole manuscript.
- The introduction should be rewritten to show the highlights and novelty of the work.
- section of drawbacks and future could be increased the quality of the manuscript.
- It is recommended to add a statement to clearly separate the current work from these similar references and also define the review period (e.g. last five years). Also, prepare statistical data (such as the number of documents, document per country) about you used references by creating databanks such as Scopus, Google Scholar, and web of science.
- Maybe at the beginning of the article, there should be a list of abbreviations?
- A review paper not only should summarize recently published works, but also should contain critical and comprehensive discussions. Therefore, check the writing for the whole manuscript. The review should not be presented by listing what have done by others.
- sections “Introduction “ and “Biosensing applications” are very poor!
Author Response
We would like to thank Reviewer 2 for his/her positive comments. The details of response please see the attachment.

Round 2
Reviewer 1 Report
n/a
Reviewer 2 Report
The paper has been improved and corresponding modifications have been conducted.I think, the current version can be considered for publication.